# Substrate multiplexed protein engineering facilitates promiscuous biocatalytic synthesis

Allwin D. McDonald ● [1,2], Peyton M. Higgins ● [1,2] & Andrew R. Buller ● [1] ✉

Enzymes with high activity are readily produced through protein engineering, but intentionally and efficiently engineering enzymes for an expanded substrate scope is a contemporary challenge. One approach to address this challenge is Substrate Multiplexed Screening (SUMS), where enzyme activity is measured on competing substrates. SUMS has long been used to rigorously quantitate native enzyme specificity, primarily for in vivo settings. SUMS has more recently found sporadic use as a protein engineering approach but has not been widely adopted by the field, despite its potential utility. Here, we develop principles of how to design and interpret SUMS assays to guide protein engineering. This rich information enables improving activity with multiple substrates simultaneously, identifies enzyme variants with altered scope, and indicates potential mutational hot-spots as sites for further engineering. These advances leverage common laboratory equipment and represent a highly accessible and customizable method for enzyme engineering.

Biocatalysts are prized for their ability to perform well-defined transformations. Unfortunately, the use of enzymes in chemical synthesis is often hampered by their small or poorly understood substrate scopes[1]. Using traditional protein engineering approaches, activity can readily be increased on a model compound[2–4]. Most engineering advances have centered on smart library design[5–8] and screening speed[9–12], and engineering campaigns using diverse approaches have, at times, led to promiscuous catalysts[2,12–14]. However, the substrate scopes of intermediates along evolutionary lineages are typically unknown. Consequently, when protein engineering does yield a catalyst with a limited scope, evolution is tediously repeated to generate activity with additional substrates[8,15–17]. Screening for activity on a single substrate necessarily overlooks mutations that are activating for substrates not included in the screen and can inadvertently lead to enzymes with high activity but narrow substrate scopes[16–18]. Methods that directly assess catalyst promiscuity would overcome this recurring pitfall and enable the development and application of biocatalysts for organic synthesis, both as single enzymes and in multi-enzyme cascade settings.

An alternative to single-substrate screening is to obtain information on catalyst promiscuity by screening with multiple substrates, either iteratively or in competition. Previously, these approaches have gone by various names including fingerprinting, substrate cocktail, multi-substrate, or multiplexed assays[18–22]. To avoid confusion as to whether substrates were assayed in separate parallel reactions or in competition, we use the term substrate multiplexed screening (SUMS) to specifically refer to methods where substrates are in direct competition (Fig. 1a). A classic application of SUMS has been for characterization of native enzyme specificity, defined as the extent to which an enzyme distinguishes between substrates[21,23,24]. In pioneering work, the Reymond group showed how careful assay design to maintain initial velocity conditions can enable the high-throughput characterization of lipase and esterase substrate specificities without measuring kinetic parameters for each individual substrate[19]. Similarly, the Kries group has advanced the use of a multiplexed assay the substrate specificity of the adenylation domains of non-ribosomal peptide synthetases[21]. In each of these cases, enzymologists take exquisite care to maintain initial velocity conditions, else the connection between product abundance and the underlying kinetic parameters is lost.

Within the broader bioengineering community, there are some scenarios where SUMS is the default mode of operation. Specifically,

[1]Department of Chemistry, University of Wisconsin–Madison, 1101 University Avenue, Madison, WI 53706, USA. [2]These authors contributed equally: Allwin D. McDonald, Peyton M. Higgins. ✉e-mail: arbuller@wisc.edu

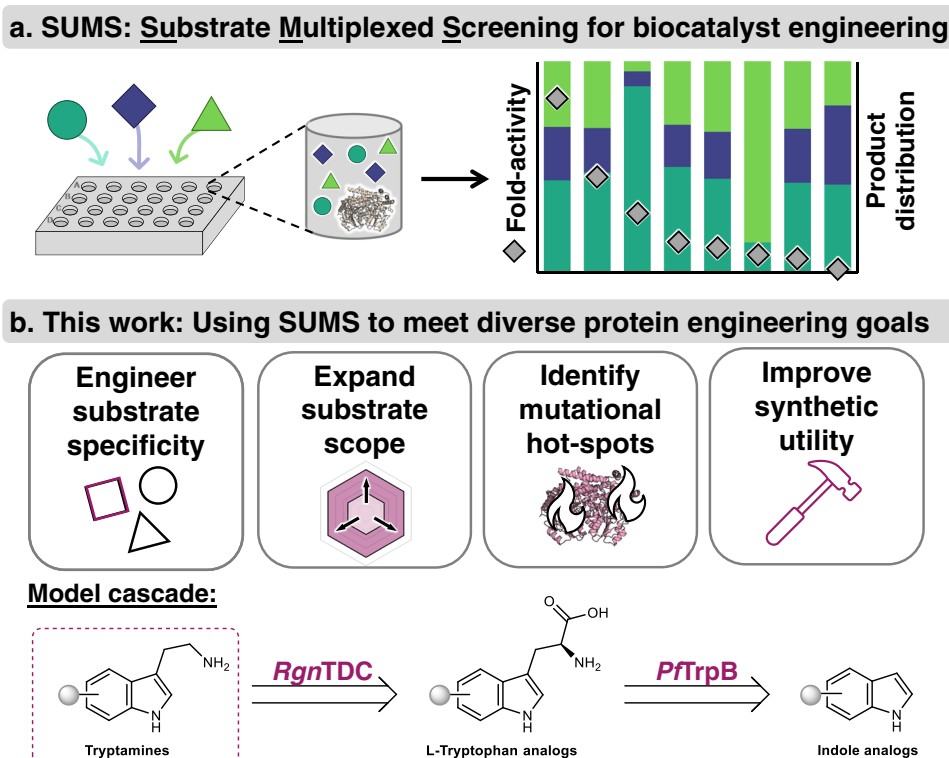

**Fig. 1 | Overview of substrate multiplexed screening (SUMS). a** In SUMS, activity is measured for multiple competing substrates simultaneously. **b** SUMS can be leveraged to address a range of goals when engineering biocatalysts. Here, we apply SUMS to a model biosynthetic cascade, forming tryptamines from indoles using *Ruminococcus gnavus* tryptophan decarboxylase (*Rgn*TDC) and *Pyrococcus furiosus* tryptophan synthase β-subunit (*Pf*TrpB).

all screens and selections that take place under in vivo conditions, or with simulated mixtures of cytosolic metabolites, necessarily incorporate substrate competition. By monitoring formation of multiple products, some groups have directly read out on catalyst specificity in order to avoid variants with undesirable off-target activity[21]. For example, the Williams group engineered the acyltransferase domain of a modular polyketide synthase to alter the substrate specificity away from the native substrate[25]. Alternatively, Weeks and Wells intentionally engineered more promiscuous subtiligase variants by screening with a peptide mixture that simulates the proteome of *E. coli* and detecting product formation using mass spectrometry (MS)[26]. Critically, in these and other cases[27–31], each enzyme's envisioned use is in an environment where substrate competition will occur.

Direct measurement of substrate specificity is a less prominent, but recurring, feature of protein engineering for organic synthesis. Although not commonly described as such, all screens with racemic substrates are, in fact, substrate multiplexed screens. Resolution of enantiomeric products requires chiral chromatography or subsequent product functionalization, which are often too cumbersome for high-throughput screening. A standard approach, then, is to first screen variants for total activity on a racemic mixture and enantiospecificity is only measured for highly active variants[32]. To circumvent this limitation, researchers have made clever use of pseudo-enantiomers to directly determine the enantioselectivity of variants in higher throughput formats[33,34]. These screening efforts focused on stereospecificity are conceptually related to, but distinct from, the more general case where researchers might screen enzymes on non-stereoisomeric substrates with the intention of monitoring changes in scope.

To date, examples of SUMS methods to expand the substrate scope of an enzyme remain comparatively rare. Jakoblinnert et al. screened carbonyl reductase variants using mixtures of three to four substrates and identified a single mutation that improved activity on four previously poor substrates. In this case, NADH depletion was measured and changes to substrate specificity were established in subsequent single substrate assays[35]. Junker et al. successfully applied a two-substrate SUMS method to simultaneously evaluate aldolase variants for changes in both activity and specificity with aldehyde and ketone-derived nucleophiles[36]. Recently, Knorrscheidt et al. demonstrated how a SUMS method using MISER-GCMS and a cocktail of three substrates could successfully identify mutations that altered the activity, specificity, and regioselectivity of an unspecific peroxygenase[37]. Given the ubiquity of SUMS in metabolic engineering and chemical biology, the success of these examples for biocatalysis, and the increasing interest in engineering enzymes for organic synthesis, it is striking that SUMS remains a specialized and uncommon approach in this field.

The transition from using a single substrate to screening with multiple substrates introduces significan and poorly understood complexities that we hypothesize have hindered wider adoption of SUMS. The kinetics of substrate competition impact assay results, as both substrates and products may act as inhibitors of the enzyme being engineered[38]. Because competing substrates exit the initial velocity regime at different rates, quantitative specificity information may be lost. Relatedly, it is not immediately apparent how well measurements in a multiplexed setting will correlate with synthetic utility for single substrate reactions. Although data from SUMS are intrinsically richer and might be leveraged in unique ways to guide engineering, data analysis and presentation additionally become more challenging as the number of potential products increases[36,37].

We set out on a systematic exploration of the We set out on a systematic exploration of the advantages and disadvantages of using SUMS to engineer enzymes for organic synthesis. We chose two enzymes for this investigation. The first enzyme is the L-tryptophan (Trp) decarboxylase from *Ruminococcus gnavus* (*Rgn*TDC), an enzyme that natively catalyzes the decarboxylation of Trp to form tryptamine

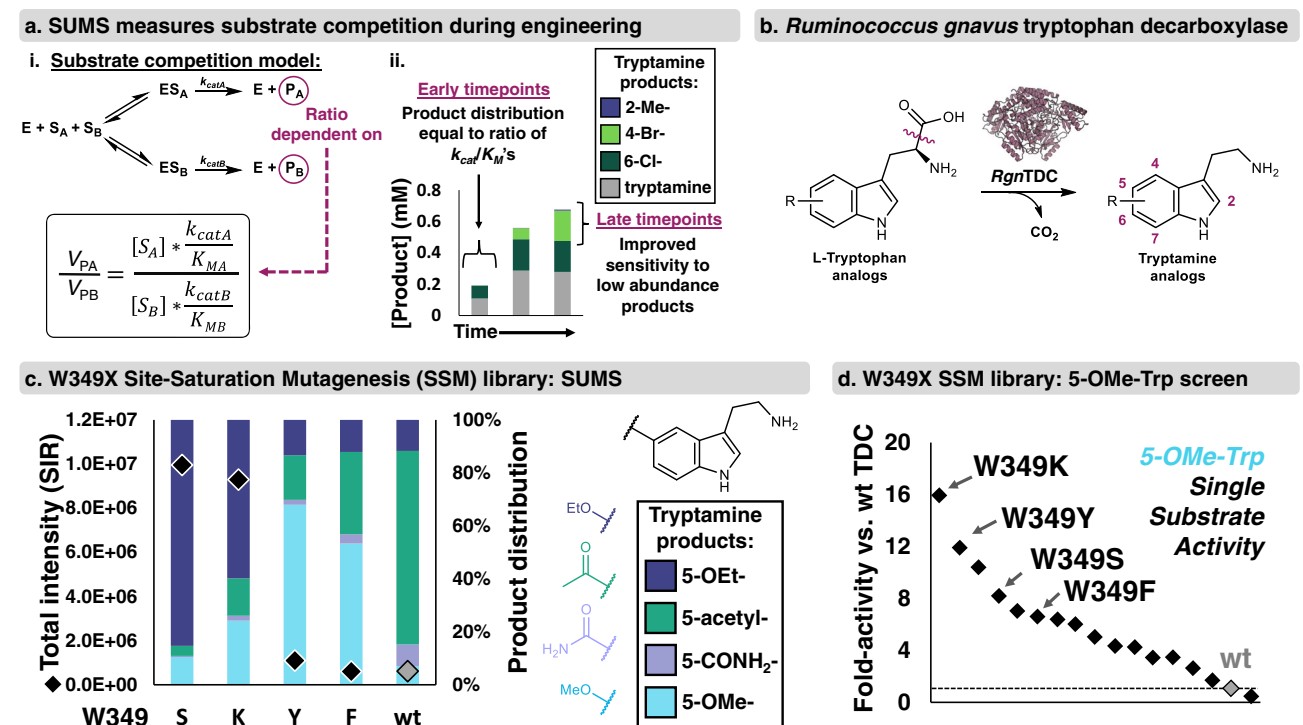

**Fig. 2 | Substrate multiplexed screening (SUMS)-based engineering of *Ruminoccus gnavus* tryptophan decarboxylase (*Rgn*TDC). a** i. Substrate competition model with equation describing relative rates of product formation. ii. Timecourse of a substrate multiplexed reaction of *Rgn*TDC with 2-Me-, 4-Br-, 6-Cl-, and Trp. Full reaction conditions found in Supplementary Fig. 42. **b** General reaction of *Rgn*TDC, with the labile bond highlighted. *R* = halo, alkyl, nitro, ether, etc. See ref. 40 for detailed scope of the wild-type enzyme. **c** SUMS results from a W349X library using a mixture of Trp substrates where *R* = 5-OEt-, 5-acetyl-, 5-CONH₂-, 5-OMe-, and 5-OMe-2-Me-. Colored bars indicate relative abundances of each product, and black diamonds indicate total intensity of single ion retention (SIR) for each product's unique m/z. No product was observed from 5-OMe-2-Me-Trp. **d** Fold-activity relative to wild-type from a single-substrate screen of the W349X library with 5-OMe-Trp corresponding to classical protein engineering techniques. Retention of function curves with full sequence analyses are shown in Supplementary Figs. 3–4.

(Fig. 1b)[39,40]. *Rgn*TDC is an exceptional decarboxylase with many Trp analogs but struggles with the highly bioactive 4- and 5-substituted substrates[40]. The second enzyme we screen is a previously-engineered β-subunit of tryptophan synthase from the thermophilic archaeon *Pyrococcus furiosus* (*Pf*TrpB^2B9, or 2B9), which catalyzes the bimolecular condensation of Ser and indole[13,16,41,42]. It has been demonstrated that 2B9 possesses a broad substrate scope, and is capable of producing diverse Trp analogs.

Here we directly compare the effectiveness of SUMS and single substrate screening for site-saturation mutagenesis and show that SUMS reveals counter-intuitive trends in substrate promiscuity. We show that SUMS can derive new information from random mutagenesis libraries in comparison to single substrate screens, further increasing the utility of random mutagenesis approaches. We combine the engineered TrpB and TDC enzymes to synthesize substituted tryptamines from L-serine (Ser) and indole. Together, these efforts show how SUMS can be used to immediately assess the substrate scope of variant libraries, form new hypotheses about enzyme function, and lead to synthetically useful enzymes for the efficient construction of bioactive molecules.

## Results

### Consideration of kinetics underlying substrate competition
Before we began screening libraries, we investigated the many variables of SUMS, such as substrate choice, relative substrate concentrations, and assay duration, that can impact the observed product profiles. To connect the SUMS output to the underlying kinetics on single substrates, we used *Rgn*TDC as a model system. For a unimolecular reaction under initial velocity conditions with equimolar substrates in competition with one another, the product abundances will be exactly proportional to the catalytic efficiencies ($k_{cat}/K_M$) of the individual reactions in isolation (Fig. 2a)[38,43,44]. As has been described, this relationship holds true even when the individual substrate concentrations exceed their $K_M$'s[38,43]. We measured traditional Michaelis-Menten parameters for *Rgn*TDC with a variety of substituted Trp analogs (Supplementary Table 1). Comparison of these data to results from multiplexed reactions showed that the ratio of the catalytic efficiencies is indeed deterministic of the product ratios (see SI discussion). As has long been appreciated in enzymology, such multiplexed activity measurements are a true measure of specificity and provide rich kinetic information about enzyme function[19]. However, these relationships are restricted to initial velocity conditions and are an incomplete measure of synthetic utility.

To capture enzyme stability effects and achieve high conversions, effective screening conditions for biocatalysis applications often utilize longer reaction times beyond the initial velocity regime. When reactions are run to higher conversion, the product profile becomes uncoupled from the Michaelis-Menten kinetics and is, instead, a heuristic readout of reactivity. Additionally, both substrates and products can inhibit enzyme activity. Often, but not always, more reactive substrates act as strong competitive inhibitors of activity on poor substrates[43,44]. Regardless, we posit that by screening on a mixture with both highly and poorly reactive substrates, we can identify catalysts that retain the ability to operate at high turnover numbers as well as identify desirable increases in activity with multiple sluggish substrates. This information is useful for organic synthesis, where enzymes would ideally react with a wide range of substrates and drive reactions to high yield. By altering the substrate composition and reaction time, screening conditions can quickly be tuned to match diverse engineering goals (Fig. 2a, see SI discussion).

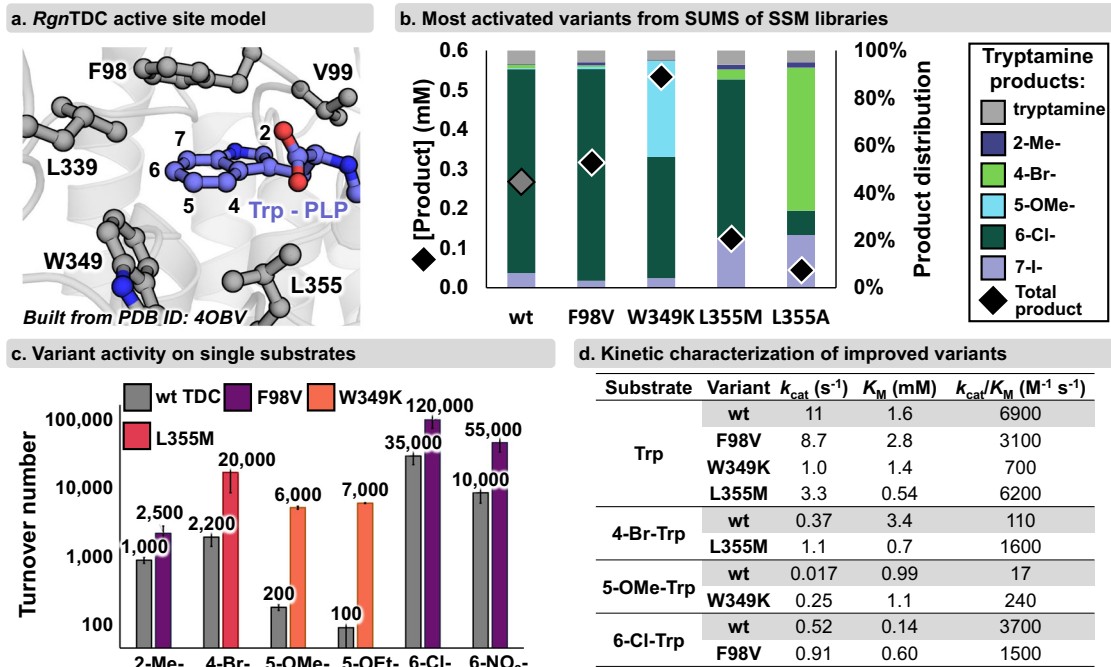

**Fig. 3 | SUMS identifies *Rgn*TDC active site mutations that improve activity for a range of substrates. a** Active site model of *Rgn*TDC (built from PDB ID: 4OBV)[39] with residues highlighted at which mutations were found that significantly altered promiscuity or improved activity. **b** Select improved variants from active site libraries. Substrate screening conditions: 0.2 mM Trp and 7-I-Trp, and 2 mM 2-Me-Trp, 4-Br-Trp, 5-OMe-Trp, and 6-Cl-Trp, 4 h, 37 °C. Colored bars indicate the relative abundances of each product, and black diamonds indicate the total product formed. Full screening results found in Supplementary

Figs. 6–S14. **c** Turnover numbers of wild-type *Rgn*TDC and the top improved variant for each substrate. Different variants are depicted by different colored bars. Turnover numbers are presented as the averages of technical triplicate measurements with standard deviation shown as a bar. **d** Michaelis-Menten parameters for wild-type *Rgn*TDC and activated variants for Trp and Trp analogs. Kinetic and turnover data were conducted in triplicate and complete data including error analysis are shown in Supplementary Tables 1 and 2 and Supplementary Figs. 17–21.

## SUMS can directly identify mutations that impact substrate promiscuity

We set out to apply a SUMS method to screen *Rgn*TDC libraries; our engineering goal was to identify mutations that either increased activity on a single substrate or on multiple Trp analogs. We began by engineering for higher *Rgn*TDC activity on 5-substituted Trp analogs, and structure-based modeling suggested the active site residue W349 forms preclusive steric interactions with these substrates (Supplementary Fig. 2). We screened a site-saturation mutagenesis (SSM) library, which exchanges the native residue for each other proteinogenic amino acid, at W349 with a mixture of five Trp substrates. For most of the substrates, we found that many mutations increased activity, and that increases in activity varied among the different substrates (Fig. 2c). The structurally conservative mutations W349Y and W349F increased activity most with 5-OMe-Trp relative to other substrates, whereas the W349S mutation had the highest activity increase with 5-OEt-Trp and produced the most total product. From this screen, W349K was identified as the most generally improved variant because it produced only slightly less 5-OEt-tryptamine than W349S and formed the most product with all other substrates.

To contrast the promiscuity information from SUMS with traditional approaches, we performed a single-substrate screen with 5-OMe-Trp on the same W349 library (Fig. 2d). As before, we found that most mutations increased activity with 5-OMe-Trp. However, there was a poor correlation between activity on 5-OMe-Trp and general activation on 5-substituted Trp analogs. Although W349K was the most activating mutation in both screens, mutations such as W349Y appeared to be highly reactive with 5-OMe-Trp but only poorly tolerated other Trp analogs. These results illustrate how SUMS can immediately identify shifts in both substrate promiscuity and activity with

no greater screening effort than would be required for a comparable single substrate screening approach.

While detailed structural analysis revealed W349 as a conspicuous site for improved activity on 5-substituted Trp analogs, such detailed hypotheses are not readily formed with all enzymes. We reasoned that SUMS could also be deployed in a setting where there is no specific hypothesis as to which residues govern activity with specific substrates. To simulate this common scenario, we screened a mixture of Trp analogs that were each substituted at a different position against a set of nine active site SSM libraries (Fig. 3a, Supplementary Fig. 5). From these screens, we found that mutation at two positions, L126 and H120, had only modest impacts on activity and promiscuity. Mutation at L336 and T356 resulted in many catalytically feeble enzymes, and the variants that retained activity had promiscuity profiles that were similar to wild-type. For the other sites, mutation caused large changes to apparent promiscuity while retaining significant catalytic activity. For example, we observed >50-fold activity increases with several enzyme-substrate pairs, such as L355M with 4-Br-Trp and F98V with 2-Me-Trp (Fig. 3b). Screening with this more diverse substrate mixture also revealed that W349K maintains high activity with non-5-substituted Trp substrates like 6-Cl-Trp. Other mutations, such as V99A and L339V, were less strongly activating for 2-Me-Trp and 4-Br-Trp but retained broad activity for substituted Trp analogs.

## Variants identified from SUMS have improved single substrate activity

An essential step during protein engineering is to validate the activity of any hits detected during library screening. Since our ultimate goal was to use engineered *Rgn*TDC variants to synthesize tryptamine analogs under single substrate conditions, we validated library hits using single substrate reactions and purified enzymes. Importantly,

although there are many confounding factors that make activity in competition distinct from activity on pure substrates, we found that turnover numbers from these single substrate reactions trended well with multiplexed screening results, with the engineered variants showing large increases in single substrate activity (Fig. 3c, Supplementary Table 2). Hence, in the same effort necessary to improve activity with one substrate, SUMS enabled the parallel engineering of *Rgn*TDC variants for improved activity with multiple challenging substrates.

### Kinetic characterization of *Rgn*TDC variants identified from SUMS

To understand the kinetic determinants of substrate promiscuity shifts for *Rgn*TDC variants, we measured Michaelis-Menten parameters for single substrates (Fig. 3d, Supplementary Table 1). All activated variants showed higher $k_{cat}$ values with their more reactive substrates when compared to wild-type. Notably, there was also significant variation in $K_M$ values for activated *Rgn*TDC variants, and such effects were difficult to rationalize from simple structural analysis. The W349K mutation, for example, accelerates decarboxylation of 5-OMe-Trp exclusively by increasing $k_{cat}$, with minimal impact to $K_M$ values (Fig. 3d). Molecular modeling indicates 4-substituted Trp analogs would form deleterious steric clashes with L355, and rational approaches to engineering would prescribe mutation to smaller sidechains. One such mutation, L355A, improved activity on 4-Br-Trp, but the L355M mutation was even more activating and had a decreased $K_M$ for both Trp and 4-Br-Trp compared to wild-type *Rgn*TDC (Fig. 3d). We highlight these unexpected findings as an advantage of interrogating active site libraries with SUMS, as such mutations could have been missed entirely by screening with the wrong pairings of substrate and mutational site.

### SUMS can identify distally activating mutations for a bimolecular reaction

To further develop SUMS for biocatalysis, we next turned our attention to improving the activity of the engineered tryptophan synthase β-subunit variant 2B9[13] on diverse indole analogs (Fig. 4a). Whereas *Rgn*TDC catalyzes a relatively simple unimolecular reaction, tryptophan synthase catalyzes a bimolecular reaction that is not well-described by simple kinetic models[45]. The ratio of the products from direct substrate competition can differ significantly from the ratio of the catalytic efficiencies measured in isolation[43]. Irrespective of the underlying kinetic phenomena, we reasoned stoichiometry could be leveraged to facilitate assay design of bimolecular reactions. By holding the invariant substrate (Ser) as the limiting reagent and providing an excess of the multiplexed reagents (indole analogs), information about specificity is maintained throughout the course of the reaction.

Because the parent enzyme, 2B9, already possesses modest activity on 4- and 5-substituted indole analogs, our engineering goal was to identify mutations that broadly increase activity with multiple indole substrates. *Rgn*TDC engineering (above) utilized active site mutagenesis, where small structural perturbations are expected to have large effects on $k_{cat}$ and $K_M$. However, residues that influence enzyme activity and substrate promiscuity can be distributed throughout the enzyme scaffold[46], and such distal mutations are known to modulate *Pf*TrpB function[13,41,47]. We, therefore, elected to screen a globally random mutagenesis library of 2B9 variants to determine whether a SUMS approach could lead to the identification of residues beyond the active site that alter either activity or substrate promiscuity.

To screen the 2B9 library, we selected a panel of commercially available indole analogs bearing substituents with diverse steric and electronic properties (see SI for in-depth discussion of assay optimization). Decades of studies have shown that most mutations to an enzyme have a neutral to deactivating impact on function[48,49]. Correspondingly, we observed that nearly all variants displayed total activity

that was either similar to or lower than 2B9 (Supplementary Figs. 23–25). A handful of variants appeared to increase overall product formation with little change in promiscuity (Fig. 4b). We purified the most activated variant, I102T, which contains a single mutation outside the active site, and found it was as good as or better than 2B9 with a variety of indole analogs under single substrate conditions (Supplementary Fig. 26). SUMS can thus achieve a traditional goal of globally random mutagenesis—identifying distal, activating mutations—while simultaneously providing insights into the substrate scope of the improved enzymes.

### Mutational hot-spots can be identified through a shift in the product profile

A unique strength of SUMS is that the promiscuity of *all* variants, activated or deactivated, is assessed, providing an additional metric by which to evaluate variants. For example, we observed a variant with lower overall activity but with a significant shift in product distribution towards 2,3-dihydroiso-L-tryptophan (DIT), which is generated by C-N bond formation with indoline (Fig. 4a). Under multiplexed screening conditions this variant, H275R, reproducibly generated more DIT than 2B9. Curiously, under single substrate conditions, H275R was *not* an activated variant and instead produced DIT more slowly than 2B9, leading us to investigate this apparent contradiction between SUMS results and activity on single substrates.

We turned to single substrate kinetic analysis with indole, *Pf*TrpB's native substrate, and indoline to probe why, in some cases, improved activity in a multiplexed screen does not translate to improved catalysis with a pure substrate. When only these two substrates are present (with indoline in a 10-fold excess), H275R disfavors Trp formation relative to 2B9, causing a prominent shift in the product ratio by impacting the relative activity with each substrate asymmetrically (Fig. 4c). Compared to 2B9, H275R shows a dramatic >100-fold decrease in $k_{cat}/K_M$ with indole but only a modest ~3-fold decrease in $k_{cat}/K_M$ with indoline (Fig. 4g, *see SI discussion*). This analysis affirmed that the discrepancy between SUMS results and single substrate activity can be resolved by considering the substrates' relative catalytic efficiencies in a multiplexed system. More importantly, the change in the H275R product profile immediately implies that the mutation impacts activity through cooperative interactions with the active site, rather than a global enzymatic property like protein stability. In other words, since variants of interest can be identified from a heuristic SUMS product profile, measurement of single substrate catalytic efficiencies is not necessary to identify sites that impact specificity.

We hypothesized that a different mutation at H275 might increase activity, rather than reduce it. This hypothesis was further motivated by the location of H275, which is a 'second-sphere' residue situated near the entrance to the enzyme's active site (Fig. 4d). We screened a SSM library at H275 with the same substrate mixture as before and observed a range of enzyme activities and product distributions (Supplementary Fig. 27). Several variants possessed activity and promiscuity similar to 2B9. Other variants resembled H275R, exhibiting an overall decrease in product formation and a shift in distribution to favor DIT. We also observed mutations that resulted in a general increase in activity across all substrates screened, with H275E displaying the largest boost (Fig. 4e). We subsequently validated that H275E has increased activity in single substrate reactions, and these improvements extend to substrates that were not present in the original screen, such as the sterically bulky nucleophile 5-OEt-indole (Fig. 4f). Notably, H275R was deactivated for all tested substrates, meaning that no single substrate screen could have identified the original H275R as a mutation of any interest. Critically, it was only by screening on a mixture of substrates and observing a shift in product distribution that the H275 site's role in substrate discrimination was identified. Hence, information from SUMS enabled use of a low-activity variant, H275R, as an intermediate to access a broadly activated enzyme, H275E.

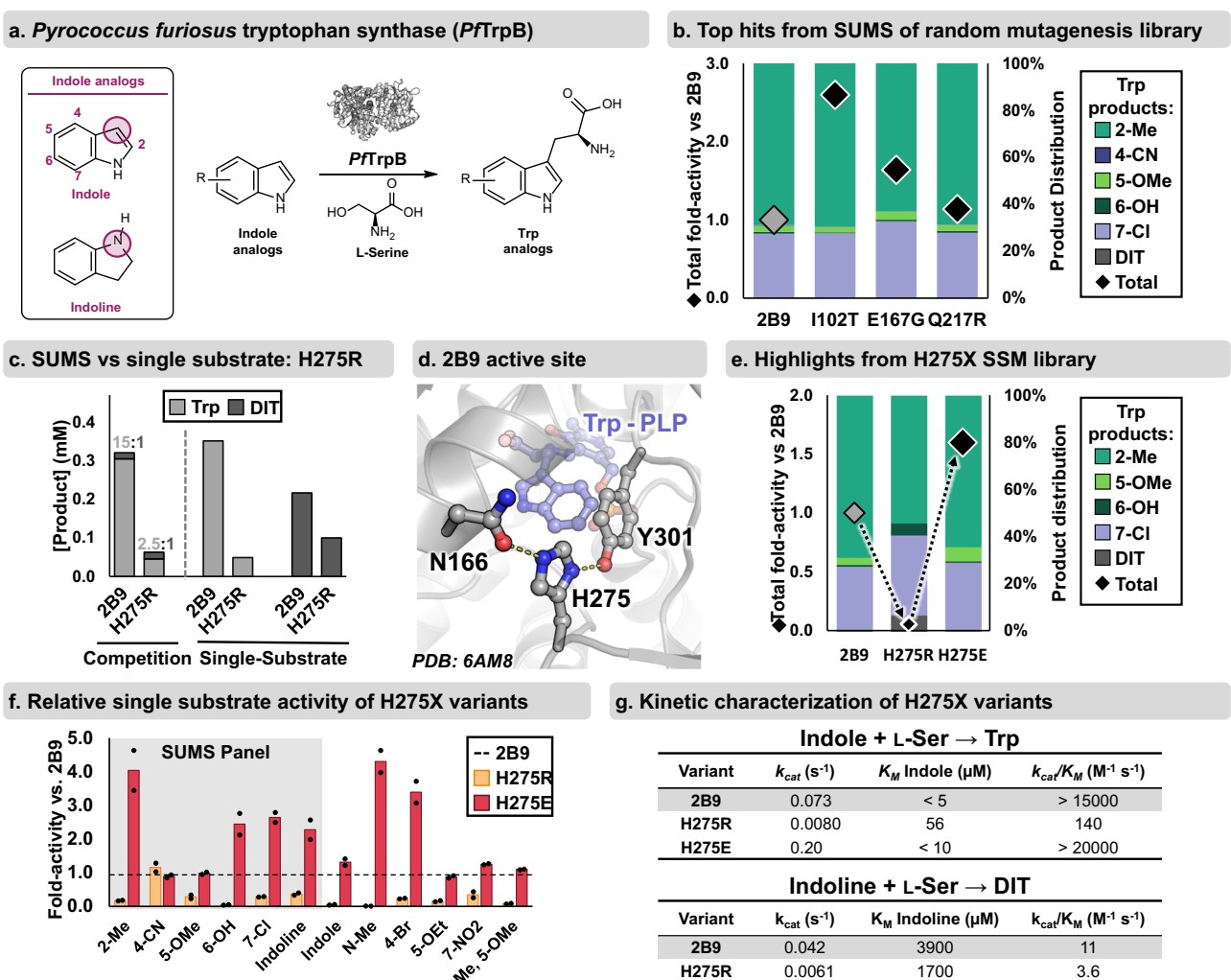

**Fig. 4 | SUMS-based engineering of the β-subunit of *Pyrococcus furiosus* tryptophan synthase (*Pf*TrpB). a** General reaction scheme for *Pf*TrpB. Indole analogs were used to screen *Pf*TrpB libraries. The nucleophilic atom is shown with a circle. *R* = halo, alkyl, nitro, ether, etc. See ref. 47 for a detailed scope of engineered TrpB enzymes. **b** SUMS results for generally activated variants detected during globally random mutagenesis library screening. Complete library results and experimental conditions are shown in Supplementary Figs. 23–25. **c** Comparison of Trp and DIT production under competition and single substrate reaction conditions, using purified 2B9 and H275R enzymes. Indoline was present at 10-fold excess (15 mM indoline, 1.5 mM indole) for both competition and single substrate reactions.

Complete duplicate data and conditions are shown in Supplementary Fig. 28. **d** H275 is a second-sphere residue that forms hydrogen bonds with neighboring residues, N166, and Y301 (PDB ID: 6AM8)[50]. **e** SUMS results for 2B9 and two variants from the H275X SSM library. 4-CN-indole was also included in reactions, but no product was observed. Complete library results are shown in Supplementary Fig. 27. **f** Product formation in single substrate reactions. Activity of H275R and H275E is shown relative to activity of 2B9 (black dashed line). **g** Michaelis–Menten parameters for 2B9, H275R, and H275E with either indole or indoline as the nucleophilic substrate. Kinetic data were measured in triplicate, and complete data including error analysis are shown in Supplementary Fig. 29.

## SUMS leads to mechanistic insights

As we found with *Rgn*TDC, SUMS-based engineering yielded thought-provoking results that raised new mechanistic questions about the causes of altered activity. Here, we were motivated to determine whether the activation afforded by H275E mimicked the same effects that were previously found in the engineering campaign that yielded 2B9[13,50].

In the absence of substrates, the PLP cofactor of *Pf*TrpB is bound to K82 as an internal aldimine, E(Ain) (Supplementary Fig. 30). We solved the structure of H275E-E(Ain) at 2.1-Å resolution, which showed a significant conformational change of a subdomain of *Pf*TrpB, called the COMM domain, relative to the parent 2B9 (Fig. 5a). Mutation at H275 disrupts a hydrogen bond network between two residues (Y181, Y301) that flank the active site and shifts the structure into the most extended-open conformational state of a TrpB observed to date[50]. Catalysis is initiated by addition of Ser, which for the parent 2B9 results in accumulation of a mixture of the Ser external aldimine, E(Aex₁), and

the electrophilic amino-acrylate intermediate, E(A-A). The activating H275E mutation shifts the ratio of intermediates to favor E(A-A), which is poised to react with a nucleophilic substrate, such as indole (Fig. 5b). Notably, the E(A-A) intermediate is also subject to a competing hydrolysis reaction. This shunt-reaction is 2.5-fold slower for H275E than 2B9 (Supplementary Fig. 31), indicating that the H275E mutation kinetically shields the reactive intermediate, affording more time for nucleophiles to react.

To determine how products fit within the active site, we solved the structure of H275E with two ligands bound, Trp and 4-Cl-Trp, at 2.39 and 2.25-Å resolution, respectively (Fig. 5c). The Trp-bound structure of H275E showed a new ligand binding pose, with the α-amine oriented for nucleophilic attack into the PLP. The structure of H275E with 4-Cl-Trp bound showed no major conformational change is required to accommodate the 4-Cl group. Instead, there is a 3.0-Å halogen bond to G298. Titrations monitored by UV-vis spectroscopy show H275E decreases the $K_D$ for Trp while simultaneously promoting a non-

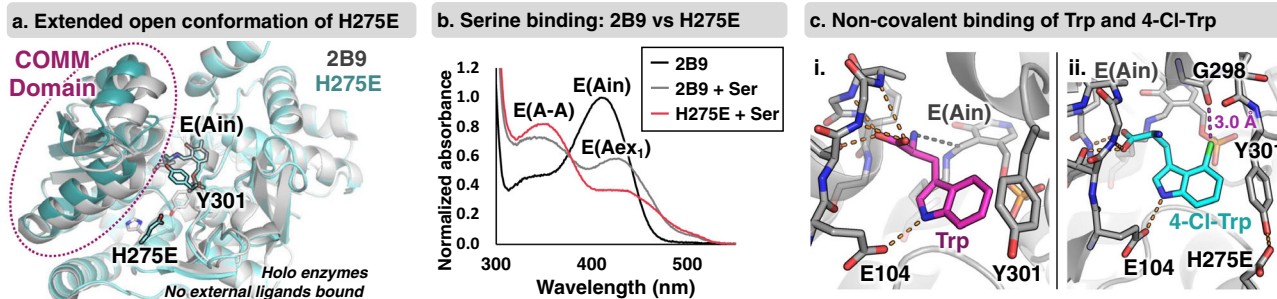

**Fig. 5 | Crystallographic and spectroscopic characterization of H275E. a** Internal aldimine structure of H275E (light blue, PDB: 7RNQ) is superimposed with the corresponding structure of 2B9 (gray, PDB: 6AM7). **b** Addition of 20 mM L-serine (Ser) to 2B9 (gray), results in two peaks corresponding to external aldimine E(Aex₁) and amino acrylate, E(A-A), intermediates. Addition of 20 mM Ser to H275E (pink)

shows a dominant peak corresponding to E(A-A). A representative enzyme-only trace is shown in black. Spectra were collected at 37 °C. **c** X-ray structures of H275E. i. Trp binding is shown in magenta from PDB: 7ROF. ii. 4-Cl-Trp binding is shown in cyan from PDB: 7RNP. Hydrogen and halogen bonds are shown in orange and purple dashes, respectively.

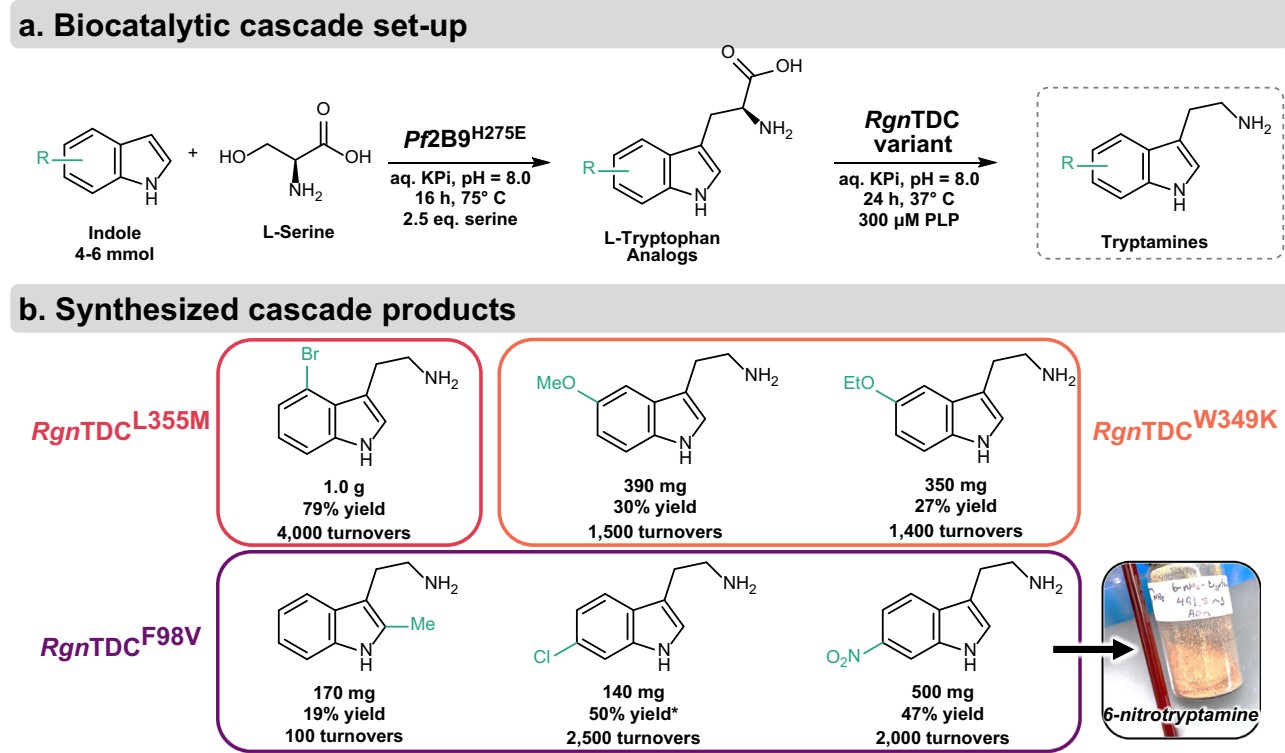

**Fig. 6 | Engineered biocatalytic cascade for synthesis of tryptamine analogs. a** Utilized biocatalytic cascade for the telescoped biosynthesis of tryptamine analogs. R = 4-Br-, 5-OMe-, 5-OEt-, 2-Me-, 6-Cl-, and 6-NO₂-. **b** Synthesized tryptamines, with the *Rgn*TDC variants used for different syntheses highlighted. Isolated yields

of tryptamine products after both steps are listed. *1.4 mmol substrate used for 6-chlorotryptamine synthesis. *Pf*TrpBᴴ²⁷⁵ᴱ loading was 0.05 mol% and *Rgn*TDC variant loading was 0.02 mol% for each reaction except for 2-methyltryptamine, which had 0.2 mol% catalyst.

covalent product binding mode (Supplementary Fig. 32). Previously, tighter binding has been associated with an increased population of covalently bound adducts[50]. Together, these data show that the H275E mutation activates *Pf*TrpB through a molecular mechanism that is distinct from previously characterized *Pf*TrpB variants, and would be difficult to identify even through contemporary, mechanism-based computational design approaches[51].

### Cascade catalysis is empowered by enzymes with complementary substrate scopes

Last, we sought to demonstrate the practical utility of the enzymes produced via SUMS. Many enzymes are more synthetically useful when employed in tandem reactions or cascades, which can overcome thermodynamic limitations and obviate the need for purification of intermediates[52]. While the use of multiple enzymes in concert can

magnify the benefits afforded by biocatalysis, catalysts must have complementary substrate scopes to synthesize a diverse set of products[52–54]. To this end, we demonstrate efficient cascade catalysis through the mmol-scale syntheses of tryptamine analogs, including 5-OMe-tryptamine and 5-OEt-tryptamine, known serotonin receptor agonists[55], and 2-Me-tryptamine and 4-Br-tryptamine, which were particularly challenging products for cascade reactions using the parent enzymes[40]. For example, 5-OMe-tryptamine was isolated in double the yield and at larger scale than our previously reported synthesis, while using only a tenth as much *Rgn*TDC catalyst. Each product was made in a telescoped biocatalytic cascade with H275E and an engineered *Rgn*TDC variant and isolated with improved yields compared to reactions with the parent enzymes (Fig. 6)[40]. Although no *Rgn*TDC variant was identified with improved activity for *all* Trp analogs, the direct assessment of substrate scope provided by SUMS

allowed us to rapidly select an improved catalyst for each tryptamine product. These reactions proceeded smoothly and afforded access to a variety of desirable tryptamines on preparative scale.

## Discussion

A central limitation to the synthetic application of many enzymes is their hard-to-predict and too-often poor substrate scopes when compared to organic methodologies[56]. Application of traditional protein engineering efficiently increases activity on model substrates, but provides no selective pressure to improve activity with a broad substrate scope. A handful of examples have demonstrated the efficacy of SUMS methods for identifying enzymes with altered promiscuity, but a lack of readily accessible explorations of SUMS for biocatalyst development has limited its general adoption. Here, we elaborated principles of SUMS assay design and data interpretation for protein engineering. We detail how factors such as relative substrate concentration, stoichiometry, and the time course of a reaction all influence the resultant product profile (see SI discussion). By thoughtfully constructing screening conditions, we showed that SUMS provides exceptional advantages when screening for increases in activity on multiple substrates and facilitates the discovery of desirable biocatalysts.

Because activity in competition is not identical to activity on isolated substrates, the information gleaned from SUMS is more than the sum of its parts. The application of SUMS here does not rely on accurate measurements of absolute substrate specificity, as screening does not necessarily take place under initial velocity conditions. Instead, we use SUMS to identify changes in the ratios of the products. Although some substrates were fully converted to product under our reaction conditions (i.e., tryptamine with *Rgn*TDC), we identified variants with retained or improved activity on these compounds in single substrate reactions. Additionally, we frequently observed mutations that appeared neutral or even deactivating with respect to one or more substrates in the reaction but were activating on another substrate in the mixture. Even when mutations result in lower activity on all substrates, an asymmetric loss in activity is direct evidence that a mutation is influencing the active site. This information would be laborious to obtain via a series of single substrate reactions but is inherent to each multiplexed measurement in a screen. Consequently, SUMS increases the utility of random mutagenesis libraries by providing intrinsically richer information about every variant. We leveraged this information to identify the *Pf*TrpB 2B9 variant H275R. This mutation lowered overall activity with each substrate and, under traditional single substrate conditions, would have been indistinguishable from mutations that destabilize the protein or reduce catalyst concentration. Instead, SUMS shows this position of the protein was a potential 'hot-spot' for mutagenesis and led to generally activating mutations at this site.

Using the additional information measurable through SUMS, researchers will be enabled to make more informed choices about which variant(s) to carry forward during directed evolution. A variant that is highly active and selective for a single substrate may be well-suited for subsequent rounds of engineering when a specific reaction is the desired goal. In contrast, mutations that are broadly activating are often more desirable for general synthetic applications. Because the selection of activating mutations has historically been made without regard to the potential impact on the scope of an enzyme, the consequences of mutational choice with respect to promiscuity are unknown. It is generally held that there is an activity-specificity tradeoff, but this hypothesis has seldom been tested by intentionally engineering for promiscuity[57]. More importantly, because catalytic perfection is not needed for an enzyme to be useful, it remains to be determined what practical limits there are to evolution for both broad scope and high catalytic efficiency. We anticipate that SUMS methods will facilitate answering these fundamental questions. More immediately, SUMS remains an easy-to-implement strategy for

biocatalyst engineering where enzymes with a broad and/or well-defined substrate scope are desired for efficient chemical synthesis.

The exploration of SUMS described here used LC-MS for detection of products, but any method that enables parallel resolution of products is compatible with SUMS. Indeed, even though LC-based screening methods tend to be slow, their versatility has led to the widespread adoption for screening libraries[2,4,9,15,37,58,59]. The substrate mixtures used here featured compounds with which the parent enzyme already had detectable baseline activity. Although it is ideal to conduct assays with rigorous standard curves for product quantification, SUMS can identify variants based on *relative* changes in product distribution. Consequently, SUMS data can be analyzed without quantitation of absolute product concentration. This feature is of practical importance, as access to products can be limited at the beginning of directed evolution campaigns when enzymatic activity is low. SUMS could even be applied with substrates that do not react with the parent enzyme, facilitating the discovery of altogether new reactions.

SUMS methods can also provide insight into whether an improvement in activity is due to changes in protein expression, for example, due to inter-well variation in a screen or a genuine change in soluble expression. If a significant shift in the SUMS product distribution is observed, this is indicative of a change in the catalytic properties of the enzyme, not protein concentration. Previously, high-throughput screening methods incorporated a normalization or tagging procedure to account for the effects of variation in protein concentration[60,61].

The work here focused on engineering enzymes with broad substrate scopes for biocatalytic applications. However, SUMS is not limited to only searching for enzymes with highly promiscuous activities; in some cases, enzymes that discriminate against particular substrates – such as different stereoisomers – are desirable. While the enzymes deployed here did not generate new stereocenters, there is no intrinsic limitation that prevents using SUMS to evolve for scope and stereoselectivity at the same time. Such multiplexed approaches for enantioselective small-molecule catalysis have seen sporadic use[20], and a similar need to identify core principles of multiplexed assay development was recently identified[62]. For these reasons, we anticipate that SUMS will become a valuable methodology for the broader catalysis community.

We provide here an exploration of how to use SUMS to engineer enzymes with improved activity on multiple compounds simultaneously. By directly assessing enzyme activity on substrates in competition, SUMS provides uniquely rich promiscuity information that has hitherto been underutilized. Importantly, just as knowledge of enzyme mechanism is not a prerequisite for the effective application of directed evolution, a priori kinetic knowledge is not required for the design and interpretation of effective multiplexed screens. These results, and the generality of the assay design principles described here, suggest the potential for SUMS to be applied to virtually any class of enzyme. Hence, the ease of implementing SUMS should minimize barriers to adoption of this budding approach.

## Methods

### Screening of *Rgn*TDC site-saturation libraries
Cell pellets were thawed and then resuspended in lysis buffer: 50 mM potassium phosphate buffer (pH = 8.0), 1 mg/mL Hen Egg White Lysozyme (GoldBio), 0.2 mg/mL DNaseI (GoldBio), 1 mM MgCl$_2$, and 300 μM pyridoxal 5′-phosphate (PLP). A volume of 600 μL lysis buffer per well was used. After 45 min of shaking at 37 °C, the resulting lysate was then spun down at 4000×$g$ to pellet cell debris. Then, 180 μL of the resulting supernatant was added to 20 μL of a substrate mixture in a separate reaction plate. Final substrate concentrations are as follows: **W349X 5-substituted Trp screen**: 2 mM each of 5-methoxy-tryptophan, 5-ethoxytryptophan, 5-methoxy-2-methyltryptophan, 5-

carboxamidotryptophan, and 5-acetyltryptophan; **W349X single-substrate screen**: 2 mM 5-methoxytryptophan; **active site site-saturation mutagenesis screens**: 2 mM each of 2-methyltryptophan, 4-bromotryptophan, 5-methoxytryptophan, and 6-chlorotryptophan; 0.2 mM 7-iodotryptophan and 0.2 mM tryptophan. Reactions were incubated at 37 °C for 4 h, quenched via addition of 150 μL 1:1 acetonitrile:1 M HCl, and centrifuged at 4000×$g$ for 10 min. 200 μL of the quenched reaction mixture supernatant was filtered into a 96-well plate for UPLC-MS analysis. Data were collected on an Acquity UHPLC with an Acquity QDA MS detector (Waters) using an Intrada Amino Acid column (Imtakt). Tryptamine product m/z ion counts were used to quantify product formation from the tryptophan reaction mixture from corresponding standard curves (Supplementary Fig. 15).

### Screening of *Pf*TrpB libraries

Lysis buffer was prepared as described above and used to resuspend cells expressing *Pf*TrpB variants. Cells were lysed for 1 hour at 37 °C then heat treated at 75 °C for 15 min. After cooling on ice, lysate was spun down at 4000×$g$ at 4 °C for 20 min. A 96-well plate was loaded with 20 μL substrate mixture (final concentration of 5 mM each 2-methylindole, 4-cyanoindole, 5-methoxyindole, 6-hydroxyindole, and indoline, plus 2.5 mM 7-chloroindole). All indole stocks were prepared in DMSO. For globally random mutagenesis library plate A, potassium phosphate buffer (50 mM, pH = 8.0, 160 μL) containing L-serine (5 mM final concentration) was added, followed by heat-treated lysate (20 μL). For subsequent plates, lysate volume was increased to 50 μL and buffer volume reduced to 130 μL. Reactions were set up such that the DMSO cosolvent comprised 10% of the final reaction volume (200 μL). Reactions were run at room temperature (25 °C) for 2.5 h and were quenched with 200 μL of acetonitrile containing 0.1 M HCl and 1 mM tryptamine (as internal standard). Plates were spun down at 4000×$g$ at 4 °C for 20 min. A 200 μL aliquot of each quenched reaction was filtered into a 96-well plate for analysis by UPLC-MS. Product formation was quantified by integration of peaks on single ion retention (SIR) channels corresponding to each expected product, normalized against the tryptamine internal standard.

### Cascade synthesis and isolation of tryptamines

4–6 mmol (1.4 mmol for 6-chloroindole) of the corresponding indole analog was added to a 1 L Erlenmeyer flask and dissolved in 20 mL MeOH. 12 mmol Ser was added, and the resulting solution was diluted up to just under 500 mL with 50 mM potassium phosphate buffer (pH = 8.0). PLP was added such that the final concentration was 300 μM. Then, H275E was added at 0.05% mol catalyst relative to the indole analog. The solution was incubated at 75 °C for 16 h. (H275E was found to be activating at 75 °C, Supplementary Fig. 33). Following UPLC-MS analysis of conversion, the solution was cooled to 37 °C, upon which *Rgn*TDC was added at 0.02–0.2% mol catalyst relative to the indole. The solutions were incubated at 37 °C for 24 h. Solutions were then evaporated down to 50–100 mL. To break emulsions, the solutions were acidified with 6 M HCl until pH < 1, 100 mL ethyl acetate (EtOAc) was added, and the resulting mixtures were centrifuged at 4000×$g$ for 10 min. These solutions were added to a separatory funnel, the aqueous layer was drained, and the organic layer removed. This was repeated twice more, with 2 mL 6 M HCl added in between extractions. Then, the aqueous layer was alkalized with 6 M NaOH until pH > 12. Tryptamine products were then extracted 3× with 150 mL EtOAc, with 2 mL 6 M NaOH added in between extractions to the aqueous layer. Organic layers were pooled, dried with sodium sulfate, filtered, and evaporated down to 5–10 mL. Solutions were transferred to 20 mL scintillation vials, evaporated to near dryness (some tryptamines were observed as liquids at 50 °C), and dried under vacuum overnight. Dried samples were weighed and submitted for ¹H and ¹³C NMR analysis.

### Reporting summary

Further information on research design is available in the Nature Research Reporting Summary linked to this article.

### Data availability

Structural data that support the findings of this study have been deposited in the Protein Data Bank with the PDB accession codes [7ROF, 7RNQ, 7RNP]. All other data that support the findings of this study are available from the corresponding author upon request. DNA and primer sequence information are available in Supplementary Table 5. Supplementary data supporting the scientific claims, supplementary figures, as well as a supplemental discussion, can be found in the Supplementary Information file.

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

## Acknowledgements

We thank professor Samuel Gellman, Professor Patrick Willoughby, Lydia Perkins, Jon Ellis, and the Buller Buddies for the helpful discussion, insight, and editorial comments. We thank Dr. Craig Bingman for assistance with crystallographic data collection. This work was supported by the Office of the Vice Chancellor for Research and Graduate Education at the University of Wisconsin-Madison with funding from the Wisconsin Alumni Research Foundation and the NIH (DP2-GM137417) to A.R.B.; and an NSF Graduate Fellowship (DGE-1747503) to P.M.H. Any opinions, findings, conclusions, or recommendations expressed in this material are those of the author(s) and do not necessarily reflect the views of the National Science Foundation. The Bruker AVANCE III-500 NMR spectrometers were supported by the Bender Fund. Use of the Advanced Photon Source was supported by the U. S. Department of Energy, Office of Science, Office of Basic Energy Sciences, under Contract No. W-31-109-Eng-38. In addition, we thank NIH Grant 1S10OD020022-1 for providing funding for the Q Extractive Plus Orbitrap used for high-resolution mass spectrometry analysis of prepared compounds.

## Author contributions

A.D.M., P.M.H., and A.R.B. designed experiments, analyzed data, and prepared the manuscript; A.D.M. and P.M.H. conducted experiments.

## Competing interests

A.R.B. is an inventor on patents for the synthetic use of modified TrpB enzymes. A.R.B and A.D.M have a patent pending describing the synthetic use of modified RgnTDC enzymes. The other authors declare no competing interests.
