## [Peer Review File · Nature Communications]

Substrate multiplexed protein engineering facilitates promiscuous biocatalytic synthesisEditorial Note: This manuscript has been previously reviewed at another journal that is not operating a transparent peer review scheme. This document only contains reviewer comments and rebuttal letters for versions considered at *Nature Communications*.

REVIEWER COMMENTS

Reviewer #1 (Remarks to the Author):

I am still fond of this manuscript and think it could be an important contribution to the field of biocatalysis and particularly protein engineering. However, as my raised points about demonstrating the broadness of the approach were not addressed, I cannot recommend publishing this work in a prestigious journal like *Nature Communications*. I am sure though, it will find a highly interested readership in more specialized journal.

Reviewer #3 (Remarks to the Author):

Buller et al present the use of SUBstrate Multiplex Screening (SUMS) as a strategy for expanding the substrate scope of a biocatalyst during iterative rounds of protein engineering/directed evolution.

SUMS has been previously reported for screening mixtures of substrates against panels of enzymes (e.g. lipases/esterases) as well as to expand the substrate scope of an unspecific peroxygenase (UPO).

The current manuscript argues that mixed-substrate screening can complement other strategies for identifying promiscuous enzymes from protein engineering/directed evolution campaigns.

The authors also argue for that benefits of SUMS include (i) reduced screening effort; (ii) identification of substrates with better conversion profiles rather than simply better initial velocities. This second point is a good one since as the authors point it is often the practical preparative scale application of the biocatalyst that is the target and hence screening under more process relevant conditions using SUMS is more pertinent.

The authors selected the enzymes TrpDC and TrpB to illustrate the benefits of SUMS and used 4-5 structurally related substrates per round of screening. Using this approach they identified a new mutation (H275R) which had not previously been reported for alternative directed evolution strategies.

The manuscript is generally well written, organised and illustrated and hence easy to read. The information is presented in a logical manner which makes it relatively straightforward to follow the line of argument.

The work appears to have been carried out with care and the data presented will help guide others who wish to adopt the SUMS approach for protein engineering.

Reviewer #4 (Remarks to the Author):

Dear Professor Buller,

Over 20 years ago, Reetz et al. described a mass spectrometry method that enables the determination of enantioselectivity of about 1000 variants per day. Their method was based on the use of "pseudo-enantiomers" where one enantiomer is isotopically labelled, making the products

distinguishable by mass spectrometry. The two isomers are mixed in a 1:1 ratio, simulating a racemic mixture, and used for screening. The ability to differentiate products by MS obviated the need for more time-consuming chromatographic analyses.[1] The approach by Reetz is conceptually similar to one used by Ma et al. to screen for hydrolase variants with improved enantioselectivity. They used a dual-colour microfluidic droplet sorter and two fluorogenic pseudo-enantiomers. Hydrolysis of the R and S enantiomers thus generated fluorescent products with different colours, enabling both total activity and selectivity to be assessed in a single measurement. This enabled them to screen millions of mutants and identify a variant with 700-fold improved enantioselectivity for (S)-profens.[2] This differs from your study in that only two substrates were used and that the substrates were fluorogenic. However, conceptually, these are not major distinctions and this suggests that the SUMS approach is not very novel. However, in response to Reviewer 2's concerns about novelty you argued that while using mixtures of substrates is not new, "leveraging mixtures of substrates for protein engineering was not well-developed". Therefore, I did a brief literature study to investigate the novelty of the SUMS approach for protein engineering. While my search was far from exhaustive, I found several papers describing the use of substrate mixtures in screening protein engineering libraries. Jakobinnert et al. used mixtures of three or four substrates to screen site-saturation mutagenesis libraries of the carbonyl reductase CPCR2. Individual products were not identified since an NADH depletion assay was used. However, starting from mixtures of substrates that are poorly converted by the wild-type enzyme, it was possible to identify mutants that increase activity on at least one of the substrates in the mixture. The hits could then be rescreened using the individual substrates.[3] Reinen et al. argued that in order to improve screening throughput, it may be useful to screen using assays in which different substrates are used in a cocktail format. They used a cocktail of six substrates to screen a library of 83 mutants of the P450 monooxygenase BM3 by UPLC-MS/MS. They demonstrated that the use of the substrate cocktail assay and chemometrics facilitates screening for both activity and diversity of product formation.[4] Junker et al. screened a small combinatorial library of fructose-6-phosphate aldolase variants using a mixture of acetone and propanal as competitive substrates. The conditions were chosen so that substrate competition facilitated the direct evaluation of both the activities and nucleophile selectivities of the variants. While they used thin layer chromatography for analysis of a small number of variants (48 mutants required screening about 200 clones), other chromatographic or mass spectrometric equipment could have been used.[5] Si et al. reported a label-free high-throughput screening method based on optically-guided MALDI-ToF MS analysis of bacterial colonies. To engineer substrate specificity, relative ion intensities of rhamnolipids were evaluated. Altered substrate specificities were reflected in mass spectra because of differences in the relative ion intensities of the products. Thousands of clones from an error-prone PCR-based random mutagenesis library could be screened using this method, allowing the identification of variants with altered selectivity.[6] Finally, Kalkreuter et al. used a multi-substrate competition assay to probe the extender unit selectivity of the wild-type Ery6TE, which forms part of the DEBS polyketide synthase. The extender unit selectivity of wild-type Ery6TE was determined in vitro using a pool of competing extender units at equimolar concentrations. This demonstrated that 58% of the native product formed, while other products ranged from 1% to 27% of the total. They then used this multi-substrate competition assay to probe a small mutant library (32 variants). They showed that some mutations led to changes in product distributions, which indicated changes in substrate selectivity.[7]

The use of substrate mixtures in protein engineering is therefore well established. I would argue that the major reason this is not done more frequently is because the approach usually requires chromatographic or mass spectrometric methods, which are lower in throughput than (less ideal) colorimetric assays. While you disagreed with the "classic perspective" of Reviewer 2, it is still true that in many cases libraries are large and high-throughput screening methods help when they are available. In reality, good high-throughput screening methods are rare and many projects involve screening smaller numbers of variants by chromatographic or mass spectrometry methods. In this sense, I do agree that the SUMS approach has value since advanced analytics are being employed anyway and it is better to get data on several compounds (and selectivity) at once. Another reason for the infrequent use of the SUMS approach might therefore be that the approach has not been clearly formulated. Therefore, your work is definitely worth publishing, but a major revision of the paper would be appropriate. I would suggest conducting a more thorough search of the literature on "multi-substrate competition assays". The core features that distinguish the SUMS approach from previously

published work should then be emphasised.

In my view, the most interesting aspect of the SUMS approach is the concept that product ratios automatically give information on specificity, regardless of absolute quantification and especially for variants that perform worse with any single substrate. Normally specific activities, which are hard (but not impossible) to determine from crude lysates in a high-throughput fashion, would be needed for each substrate to achieve the same comparison. More importantly, screens that focus on increased product formation would fail to identify sites that influence specificity. For example, the you show that the H275R variant was deactivated for all the substrates tested, meaning that no single-substrate assay "could have identified the original H275R as a mutation of any interest". However, this interpretation is inaccurate. First, other mutations at that site, like the broadly activated H275E, could have been found using single-substrate screens. Second, loss of activity does identify a site as potentially interesting, which is why alanine scanning is still often employed.

You discuss an apparent contradiction between the results of a SUMS screen and single-substrate assays. Under multiplexed conditions the H275R variant reproducibly generated more DIT than 2B9. The reader is referred to Figure 4c, where unfortunately it is hard to judge the absolute amounts of product formed from the plotted data. Given the importance of this data, it would be good to present the numbers. The legend refers the reader to Figure S28, where 2B9 is compared to H275E, not H275R (this mistake is not made in the legend). I would have appreciated the numbers here, perhaps this can be added as a supporting table. The counterintuitive observation you are trying to address is that under single-substrate conditions, the H275R variant produces less, not more, DIT than the wild-type. Therefore, the H275R variant produces more DIT than the wild-type under multiple-substrate conditions, and less DIT than the wild-type under single-substrate conditions. You explain this by arguing that the formation of DIT by H275R is less sensitive to inhibition by indole than is the case for wild-type 2B9. I think this discussion is excessively complex. A key concept in the paper is that, for a "unimolecular reaction under initial velocity conditions with equimolar substrates in competition with one another, the product abundances will be exactly proportional to the catalytic efficiencies (k_{cat}/K_M) of the individual reactions in isolation" (lines 106-108 and Figure 2). Looking at the data in Figure S29, the K_{cat}/K_M ratios (H275R/2B9) are ~1:100 for indole and ~1:3 for indoline. The K_{cat}/K_M ratios (indoline:indole) are ~1:283 for 2B9 and ~1:8 for H275R. It is therefore, based on a key premise of the paper, not so strange that the single and multi-substrate assays give different results. I think reference to Figure S27 would help clarify things a lot. The screening conditions involved 5 mM each of 2-Me-indole, 4-CN-indole, 5-OMe-indole, 6-OH-indole, and indoline, and 2.5 mM 7-Cl-indole, with 5 mM of serine, giving a 5.5-fold excess of the indole analogues. As can be seen in Figure S27, under these serine-limiting conditions, the wild-type does not seem to form any DIT, while several variants, including H275R, do. If one substrate reacts significantly faster, using up the available serine, one would expect less of a slower-forming product to be produced. If serine was used in excess, the result may have been different. I think this is in agreement with your views (see lines 196-198) and apologise if I misinterpreted this. It therefore seems like an unnecessary complication to make so much of a result that is not fundamentally confusing in the first place. Further, while two substrates may be seen as competitive inhibitors of each other, they are still both substrates and it is more helpful to discuss them as better and worse substrates than as inhibitors. If they are discussed as inhibitors to such an extent (including a supplementary discussion) then it would seem appropriate to confirm this by determining inhibition constants. Mutants may also be faster because of reduced product inhibition (for example, see Huerta et al. [8]) and the native product of an enzyme may inhibit promiscuous reactions to varying extents, depending on the substrate (this inhibition also does not need to be competitive). Therefore, given the significant discussion of kinetics and substrate inhibition, it seems inappropriate that product inhibition was not discussed. This, too, could be experimentally investigated. Overall I would be uninterested in that experiment and would not insist that you perform such experiments. It would be easier, and clearer to the reader, to simply tone down the discussion of kinetics since sufficient data is not available.

I enjoyed reading your paper and sincerely hope my comments will make a positive contribution to your work.

Best wishes,
Chris Badenhorst

References

- [1] M. T. Reetz, M. H. Becker, H.-W. Klein, D. Stöckigt, *Angew. Chem., Int. Ed.* 1999, 38, 1758-1761.
- [2] F. Ma, M. T. Chung, Y. Yao, R. Nidetz, L. M. Lee, A. P. Liu, Y. Feng, K. Kurabayashi, G. Y. Yang, *Nat. Commun.* 2018, 9, 1030.
- [3] A. Jakoblinnert, J. Wachtmeister, L. Schukur, A. V. Shivange, M. Bocola, M. B. Ansorge-Schumacher, U. Schwaneberg, *Protein Eng., Des. Sel.* 2013, 26, 291-298.
- [4] J. Reinen, G. Postma, C. Tump, T. Bloemberg, J. Engel, N. P. Vermeulen, J. N. Commandeur, M. Honing, *Anal. Bioanal. Chem.* 2016, 408, 1425-1443.
- [5] S. Junker, R. Roldan, H. J. Joosten, P. Clapes, W. D. Fessner, *Angew. Chem., Int. Ed.* 2018, 57, 10153-10157.
- [6] T. Si, B. Li, T. J. Comi, Y. Wu, P. Hu, Y. Wu, Y. Min, D. A. Mitchell, H. Zhao, J. V. Sweedler, *J. Am. Chem. Soc.* 2017, 139, 12466-12473.
- [7] E. Kalkreuter, K. S. Bingham, A. M. Keeler, A. N. Lowell, J. J. Schmidt, D. H. Sherman, G. J. Williams, *Nat. Commun.* 2021, 12, 2193.
- [8] C. Huerta, N. V. Grishin, H. Zhang, *Biochemistry* 2013, 52, 3615-3617.

REVIEWER COMMENTS

Reviewer #1 (Remarks to the Author):

I am still fond of this manuscript and think it could be an important contribution to the field of biocatalysis and particularly protein engineering. However, as my raised points about demonstrating the broadness of the approach were not addressed, I cannot recommend publishing this work in a prestigious journal like Nature Communications. I am sure though, it will find a highly interested readership in more specialized journal.

We thank Reviewer #1 for revisiting our manuscript and their recognition that it will make an important contribution. We made substantial edits to our introduction that more clearly place our work in the context of past SUMS efforts. This framing makes it clearer how a discussion of general elements relating to assay design and interpretation is needed for this powerful approach to be adopted by the wider catalysis community.

Reviewer #3 (Remarks to the Author):

Buller et al present the use of SUBstrate Multiplex Screening (SUMS) as a strategy for expanding the substrate scope of a biocatalyst during iterative rounds of protein engineering/directed evolution.

SUMS has been previously reported for screening mixtures of substrates against panels of enzymes (e.g. lipases/esterases) as well as to expand the substrate scope of an unspecific peroxygenase (UPO).

The current manuscript argues that mixed-substrate screening can complement other strategies for identifying promiscuous enzymes from protein engineering/directed evolution campaigns.

The authors also argue for that benefits of SUMS include (i) reduced screening effort; (ii) identification of substrates with better conversion profiles rather than simply better initial velocities. This second point is a good one since as the authors point it is often the practical preparative scale application of the biocatalyst that is the target and hence screening under more process relevant conditions using SUMS is more pertinent.

The authors selected the enzymes TrpDC and TrpB to illustrate the benefits of SUMS and used 4-5 structurally related substrates per round of screening. Using this approach they identified a new mutation (H275R) which had not previously been reported for alternative directed evolution strategies.

The manuscript is generally well written, organised and illustrated and hence easy to read. The information is presented in a logical manner which makes it relatively straightforward to follow the line of argument.

The work appears to have been carried out with care and the data presented will help guide others who wish to adopt the SUMS approach for protein engineering.

We thank Reviewer #3 for their thoughtful comments and hope they will agree that our revised manuscript will even *more* effectively draw in other researchers with an interest in catalysis and promiscuity to adapt SUMS methods to their own research.

Reviewer #4 (Remarks to the Author):

The central question raised by Rev. 4 was: how novel is the work here? This is an important question. Rev. 4 identified several additional examples of researchers screening on mixtures and asked how these studies frame our work. We have addressed this question in the form of a fully revised introduction. Most of the examples raised by Rev. 4 fall under the broad category of engineering for applications where enzymes must distinguish between two or more substrates in order to be useful. Multiplexed approaches have long been used in this arena. For biocatalysis, however, the vast majority of researchers screen on single substrates, even when multiplexing would provide significant advantages. The papers identified by Rev. 4 show SUMS can be effective for biocatalysis but, crucially, provide no explanation about how to use SUMS more generally or what benefits the approach brings. We agree with Rev. 4's assessment that a "... reason for the infrequent use of the SUMS approach might therefore be that *the approach has not been clearly formulated.*" By exploring the above concepts in detail, we provide the much-needed 'clear formulation'. Further, we identify the (many) core features of the SUMS approach that were wholly missed by past studies. Indeed, a litany of seemingly basic questions about SUMS have not yet been addressed. Through new contributions in our work we:

- Explore the relationship between traditional kinetic parameters (k_{cat}/K_M) and SUMS output and explain why such information is not necessary for effective assay design.
- Describe how assay conditions (time, concentration, etc.) alter the resulting product profile.
- Clarify additional considerations unique to specificity of bimolecular reactions.
- Provide the first direct comparison of single substrate screening and SUMS to analyze a site-saturation mutagenesis library.
- Sequence entire libraries (not just 'winners') to show that seemingly conservative mutations may cause large changes in promiscuity.
- Provide an example of how to identify a mutational hotspot based entirely on changes in promiscuity from a universally de-activated variant.
- Last, we show SUMS led to a universally activating mutation that would be challenging to find through even the most cutting-edge mechanism-based computational design, because the mutation alters the mechanism in a totally new and unexpected way.

Further point-by-point responses are below.

Dear Professor Buller,

Over 20 years ago, Reetz et al. described a mass spectrometry method that enables the determination of enantioselectivity of about 1000 variants per day. Their method was based on the use of "pseudo-enantiomers" where one enantiomer is isotopically labelled, making the products distinguishable by mass spectrometry. The two isomers are mixed in a 1:1 ratio, simulating a racemic mixture, and used for

screening. The ability to differentiate products by MS obviated the need for more time-consuming chromatographic analyses.[1] The approach by Reetz is conceptually similar to one used by Ma et al. to screen for hydrolase variants with improved enantioselectivity. They used a dual-colour microfluidic droplet sorter and two fluorogenic pseudo-enantiomers. Hydrolysis of the R and S enantiomers thus generated fluorescent products with different colours, enabling both total activity and selectivity to be assessed in a single measurement. This enabled them to screen millions of mutants and identify a variant with 700-fold improved enantioselectivity for (S)-profens.[2] This differs from your study in that only two substrates were used and that the substrates were fluorogenic. However, conceptually, these are not major distinctions and this suggests that the SUMS approach is not very novel. However, in response to Reviewer 2's concerns about novelty you argued that while using mixtures of substrates is not new, "leveraging mixtures of substrates for protein engineering was not well-developed".

Reviewer #4's expert knowledge of the literature was very helpful in revising our manuscript! We have added a paragraph to the introduction that clearly spells out how screening with racemic (or pseudo-racemic) compounds is intrinsically multiplexed. However, the more general case of using mixtures to examine scope for synthesis is still severely underdeveloped, as we will explain in further detail below.

Therefore, I did a brief literature study to investigate the novelty of the SUMS approach for protein engineering. While my search was far from exhaustive, I found several papers describing the use of substrate mixtures in screening protein engineering libraries. Jakoblinert et al. used mixtures of three or four substrates to screen site-saturation mutagenesis libraries of the carbonyl reductase CPCR2. Individual products were not identified since an NADH depletion assay was used. However, starting from mixtures of substrates that are poorly converted by the wild-type enzyme, it was possible to identify mutants that increase activity on at least one of the substrates in the mixture. The hits could then be rescreened using the individual substrates.[3] Reinen et al. argued that in order to improve screening throughput, it may be useful to screen using assays in which different substrates are used in a cocktail format. They used a cocktail of six substrates to screen a library of 83 mutants of the P450 monooxygenase BM3 by UPLC-MS/MS. They demonstrated that the use of the substrate cocktail assay and chemometrics facilitates screening for both activity and diversity of product formation.[4] Junker et al. screened a small combinatorial library of fructose-6-phosphate aldolase variants using a mixture of acetone and propanal as competitive substrates. The conditions were chosen so that substrate competition facilitated the direct evaluation of both the activities and nucleophile selectivities of the variants. While they used thin layer chromatography for analysis of a small number of variants (48 mutants required screening about 200 clones), other chromatographic or mass spectrometric equipment could have been used.[5] Si et al. reported a label-free high-throughput screening method based on optically-guided MALDI-ToF MS analysis of bacterial colonies. To engineer substrate specificity, relative ion intensities of rhamnolipids were evaluated. Altered substrate specificities were reflected in mass spectra because of differences in the relative ion intensities of the products. Thousands of clones from an error-prone PCR-based random mutagenesis library could be screened using this method, allowing the identification of variants with altered selectivity.[6] Finally, Kalkreuter et al. used a multi-substrate competition assay to probe the extender unit selectivity of the wild-type Ery6TE, which forms part of the DEBS polyketide synthase. The extender unit selectivity of wild-type Ery6TE was determined

in vitro using a pool of competing extender units at equimolar concentrations. This demonstrated that 58% of the native product formed, while other products ranged from 1% to 27% of the total. They then used this multi-substrate competition assay to probe a small mutant library (32 variants). They showed that some mutations led to changes in product distributions, which indicated changes in substrate selectivity.[7]

Reviewer #4's literature search is very appreciated! We have incorporated these citations, along with additional ones we found, in a revised introduction to our manuscript. Detailed evaluation of these papers shows that the majority of the studies were conducted with an eye towards understanding/improving specificity when the target application is under competition conditions. In the cases we have been able to find where multiplexed screening is used in combination with protein engineering, the focus of the studies is squarely on the systems that they are engineering and not on understanding the strengths and limitations of multiplexing itself.

The use of substrate mixtures in protein engineering is therefore well established. I would argue that the major reason this is not done more frequently is because the approach usually requires chromatographic or mass spectrometric methods, which are lower in throughput than (less ideal) colorimetric assays. While you disagreed with the "classic perspective" of Reviewer 2, it is still true that in many cases libraries are large and high-throughput screening methods help when they are available. In reality, good high-throughput screening methods are rare and many projects involve screening smaller numbers of variants by chromatographic or mass spectrometry methods.

It is refreshing to read a review that presents a balanced perspective on a difficult question. If one looks at most of the high-profile biocatalysis papers of the past decade, there is a clear shift towards using LC, GC, or MS based methods that many would not have predicted 15-20 years ago. As Rev. 4 notes, good HT methods are rare! More specifically, there are more biocatalysis problems than there are HT methods.

In this sense, I do agree that the SUMS approach has value since advanced analytics are being employed anyway and it is better to get data on several compounds (and selectivity) at once. Another reason for the infrequent use of the SUMS approach might therefore be that the approach has not been clearly formulated.

We fully agree with Rev. 4's assessment. Their summary that 'the approach has not been clearly formulated' is precisely what has motivated our work. It is not enough for there to be a scant handful of cases where someone uses an approach that is new to the field. SUMS must be put in context for researchers to be able to assess whether the strategy is worth it for their particular problem. As we have spoken with folks the past few months, often their answer has been 'yes', that a multiplexing approach offers significant potential benefits at few added costs.

Therefore, your work is definitely worth publishing, but a major revision of the paper would be appropriate. I would suggest conducting a more thorough search of the literature on "multi-substrate

competition assays". The core features that distinguish the SUMS approach from previously published work should then be emphasised.

We found these comments especially informative in revising the introduction and discussion sections of our manuscript. As discussed above, the scientific underpinnings of how to set up and interpret multiplexed data and how this approach compares to single substrate screening is still altogether lacking. Therefore, this manuscript fills a real gap in the literature. Previous reports have shown success using multiplexing approaches without explaining how the method works, which adds to the urgency that this knowledge gap be filled.

In my view, the most interesting aspect of the SUMS approach is the concept that product ratios automatically give information on specificity, regardless of absolute quantification and especially for variants that perform worse with any single substrate. Normally specific activities, which are hard (but not impossible) to determine from crude lysates in a high-throughput fashion, would be needed for each substrate to achieve the same comparison.

We were likewise drawn to multiplexing for just this reason: it's simple to do.

More importantly, screens that focus on increased product formation would fail to identify sites that influence specificity. For example, the you show that the H275R variant was deactivated for all the substrates tested, meaning that no single-substrate assay "could have identified the original H275R as a mutation of any interest". However, this interpretation is inaccurate. First, other mutations at that site, like the broadly activated H275E, could have been found using single-substrate screens.

Here, we suspect that Rev. 4 has mis-read our assertion. We are writing specifically about H275R as a mutation that decreases activity while shifting specificity. Because sequence space is much larger than our ability to screen it, it is of independent merit to learn how to extract more information from the vast portion of sequence space that is lower in activity.

On a separate, more technical level, H275E would be unlikely to appear in an error prone mutagenesis library, since this mutation would require two base pair changes rather than one. Interestingly, the H275 site-saturation mutagenesis library revealed additional variants (such as H275W or H275P) with similar product profiles to H275R, indicating that H275R was not the only potential variant that could have indicated this position was a 'hot spot'. How common such mutations are, no one knows. We as a community have been totally blind to these effects when we screen on single substrates.

Second, loss of activity does identify a site as potentially interesting, which is why alanine scanning is still often employed.

We do not make specific assertions about the strengths and weaknesses of alanine scanning. In theory, yes, SUMS would add make this classic approach even more informative. However, it is worth considering that H275A did *not* display a significant shift in product profile from the H275 SSM screen, highlighting the advantages of more agnostic screening strategies.

You discuss an apparent contradiction between the results of a SUMS screen and single-substrate assays. Under multiplexed conditions the H275R variant reproducibly generated more DIT than 2B9. The reader is referred to Figure 4c, where unfortunately it is hard to judge the absolute amounts of product formed from the plotted data. Given the importance of this data, it would be good to present the numbers. The legend refers the reader to Figure S28, where 2B9 is compared to H275E, not H275R (this mistake is not made in the legend). I would have appreciated the numbers here, perhaps this can be added as a supporting table.

We thank Rev. 4 for their attention to detail. The typographical error in Figure S28 has been corrected to reflect the correct variants (2B9 and H275R). In addition, ratios of the observed Trp:DIT products have been added to Fig 4c in the main text to more clearly show the shift in product profile from H275R.

The counterintuitive observation you are trying to address is that under single-substrate conditions, the H275R variant produces less, not more, DIT than the wild-type. Therefore, the H275R variant produces more DIT than the wild-type under multiple-substrate conditions, and less DIT than the wild-type under single-substrate conditions. You explain this by arguing that the formation of DIT by H275R is less sensitive to inhibition by indole than is the case for wild-type 2B9. I think this discussion is excessively complex. A key concept in the paper is that, for a "unimolecular reaction under initial velocity conditions with equimolar substrates in competition with one another, the product abundances will be exactly proportional to the catalytic efficiencies (k_{cat}/K_M) of the individual reactions in isolation" (lines 106-108 and Figure 2). Looking at the data in Figure S29, the K_{cat}/K_M ratios (H275R/2B9) are ~1:100 for indole and ~1:3 for indoline. The K_{cat}/K_M ratios (indoline:indole) are ~1:283 for 2B9 and ~1:8 for H275R. It is therefore, based on a key premise of the paper, not so strange that the single and multi-substrate assays give different results.

We are heartened that Rev. 4 thinks this phenomenon is sufficiently simple. Most people that we have spoken to find the results rather counter-intuitive on their face. More importantly, the take-home message from this analysis was that detailed kinetic investigations are *not* needed to identify a variant as worthy of follow-up. We have therefore simplified our discussion of this particular phenomenon in the paper. Discussion of "repression of inhibition" has been moved to our SI Discussion, where it will be available for any readers who are curious to learn more about our interpretation of results.

I think reference to Figure S27 would help clarify things a lot. The screening conditions involved 5 mM each of 2-Me-indole, 4-CN-indole, 5-OMe-indole, 6-OH-indole, and indoline, and 2.5 mM 7-Cl-indole, with 5 mM of serine, giving a 5.5-fold excess of the indole analogues. As can be seen in Figure S27, under these serine-limiting conditions, the wild-type does not seem to form any DIT, while several variants, including H275R, do. If one substrate reacts significantly faster, using up the available serine, one would expect less of a slower-forming product to be produced. If serine was used in excess, the result may have been different. I think this is in agreement with your views (see lines 196-198) and

apologise if I misinterpreted this. It therefore seems like an unnecessary complication to make so much of a result that is not fundamentally confusing in the first place.

Rev. 4 is correct, their analysis is in agreement with our views. We have generally found that much of the SUMS process is 'not fundamentally confusing.' Nevertheless, as the goal of our paper is to present a cogent and thorough discussion of possible reaction outcomes, we would consider our job well-done if a reader finished and felt the same; we aim to demonstrate that SUMS isn't as complicated as it may seem!

Further, while two substrates may be seen as competitive inhibitors of each other, they are still both substrates and it is more helpful to discuss them as better and worse substrates than as inhibitors. If they are discussed as inhibitors to such an extent (including a supplementary discussion) then it would seem appropriate to confirm this by determining inhibition constants. Mutants may also be faster because of reduced product inhibition (for example, see Huerta et al. [8]) and the native product of an enzyme may inhibit promiscuous reactions to varying extents, depending on the substrate (this inhibition also does not need to be competitive). Therefore, given the significant discussion of kinetics and substrate inhibition, it seems inappropriate that product inhibition was not discussed. This, too, could be experimentally investigated. Overall I would be uninterested in that experiment and would not insist that you perform such experiments. It would be easier, and clearer to the reader, to simply tone down the discussion of kinetics since sufficient data is not available.

We have followed Reviewer #4's suggestion and simplified discussion of kinetics in our manuscript. We agree that discussion of product inhibition is not needed for this study. Such effects are equally prevalent (or absent) in single substrate screens and researchers are already accustomed to them. We have shifted the focus of the manuscript as a whole to better place our SUMS manuscript in the context of other multiplexed screening endeavors and to emphasize practical considerations for SUMS implementation, such as assay design and data interpretation.

I enjoyed reading your paper and sincerely hope my comments will make a positive contribution to your work.

We are deeply grateful to Reviewer #4 for their insights and hope they will agree that our revised manuscript better meets its goal of laying out the fundamentals of implementing SUMS for protein engineering to make this useful method more approachable to the wide audience of protein engineers and catalysis researchers.

Best wishes,
Chris Badenhorst

When this work is published, we hope many researchers will be empowered to take advantage of the unique benefits that SUMS may bring to tackling diverse problems. During the time that our manuscript has been in review, a distinct but thematically similar [paper](https://doi.org/10.26434/chemrxiv-2022-9w88q) appeared on a pre-print server from Prof. Eric Jacobsen's group (10.26434/chemrxiv-2022-9w88q). They try to address the problem of

developing small-molecule catalysts that are selective and have a wide scope. That paper is appearing in 2022 precisely because the question of how to effectively develop promiscuous catalysts for synthesis remains contemporary and unsolved. We include this citation in our Discussion, which adds to the urgency to begin battle-testing these new screening approaches. We have been using SUMS in our group to great effect for several years now and we anticipate this manuscript will be a foundational text that makes multiplexed screening approachable for the catalysis community.

Last, I have seldom received such a thorough and thoughtful peer review, and my team and I are grateful.

Sincerely, Andrew

References

- [1] M. T. Reetz, M. H. Becker, H.-W. Klein, D. Stöckigt, *Angew. Chem., Int. Ed.* 1999, 38, 1758-1761.
- [2] F. Ma, M. T. Chung, Y. Yao, R. Nidetz, L. M. Lee, A. P. Liu, Y. Feng, K. Kurabayashi, G. Y. Yang, *Nat. Commun.* 2018, 9, 1030.
- [3] A. Jakoblinnert, J. Wachtmeister, L. Schukur, A. V. Shivange, M. Bocola, M. B. Ansorge-Schumacher, U. Schwaneberg, *Protein Eng., Des. Sel.* 2013, 26, 291-298.
- [4] J. Reinen, G. Postma, C. Tump, T. Bloemberg, J. Engel, N. P. Vermeulen, J. N. Commandeur, M. Honing, *Anal. Bioanal. Chem.* 2016, 408, 1425-1443.
- [5] S. Junker, R. Roldan, H. J. Joosten, P. Clapes, W. D. Fessner, *Angew. Chem., Int. Ed.* 2018, 57, 10153-10157.
- [6] T. Si, B. Li, T. J. Comi, Y. Wu, P. Hu, Y. Wu, Y. Min, D. A. Mitchell, H. Zhao, J. V. Sweedler, *J. Am. Chem. Soc.* 2017, 139, 12466-12473.
- [7] E. Kalkreuter, K. S. Bingham, A. M. Keeler, A. N. Lowell, J. J. Schmidt, D. H. Sherman, G. J. Williams, *Nat. Commun.* 2021, 12, 2193.
- [8] C. Huerta, N. V. Grishin, H. Zhang, *Biochemistry* 2013, 52, 3615-3617.

REVIEWER COMMENTS

Reviewer #4 (Remarks to the Author):

I am satisfied with the revised manuscript. As was clear from my first review of this manuscript, I was initially sceptical. However, over the past several weeks I have found the concept of this paper frequently coming up while thinking about diverse aspects of high-throughput screening. While seemingly simple at first, I believe that this concept has the potential to significantly increase the amount of information extracted from variant libraries and I still think that perhaps it would be more frequently employed if the approach were more clearly formulated. I think this manuscript makes a good case for the SUMS approach and I hope to see much more of this in the future. I still have a few minor comments that the authors might consider before publication if the manuscript is accepted.

Line 72: "with a peptide mixture that simulating" should be "with a peptide mixture that simulates"

Lines 180-181: "smaller W349S mutation" sounds strange

Lines 220-221: Be more explicit about how incubation time affects product ratios, perhaps referring to Figure S42.

Lines 252-253: The authors could emphasise that this makes screening random mutagenesis libraries rather unrewarding and therefore decreases motivation to do so. If SUMS dramatically increases the chances of finding useful information (e.g., identifying residues that are important to substrate specificity even if overall activity is decreased) then screening random libraries would become much more interesting, even in cases where high-throughput methods are not available.

Line 370: This is a good place to remind the reader that it is usually critical that reactions do not go to completion.

Lines 372-374: The authors could emphasise how much more work it would require to get the same information without substrate mixtures (i.e., k_{cat}/K_M would need to be estimated for the individual substrates, merely running the reactions separately would not give the same information).

Lines 398-400: I think an even bigger problem with high-throughput screening is that there is significant inter-well variation in protein expression level, even for the same variant. This means that even if absolute product concentration is accurately quantified, no information on changes in specific activity are obtained. This could be addressed by normalisation strategies using fusion tags for quantification (split-GFP, split-luciferase, S-tag, GST-tag, etc.) but throughput is decreased considerably due to the complexity and time required for these procedures. Worse, some proteins do not tolerate tags at either end, meaning that normalisation in crude lysates is essentially impossible. Overall, this makes high-throughput screening nonquantitative, which in turn severely limits its utility. This problem is making it very difficult to get high-quality data for machine learning projects, to mention just one example. The ability to get useful information on specificity without the need for normalisation based on expression levels is therefore one of the most exciting benefits of the SUMS approach and I believe the authors would make the significance of their work much more obvious by emphasising this aspect.

Line 557: Reference page numbers are wrong: 2193

Line 573: Reference information is wrong: 1999, Pages 1758-1761

Line 575: Page numbers are wrong: 1030

General comments:

- Was Reinen et al., (Anal. Bioanal. Chem., 2016, 408, 1425-1443) left out intentionally? This is one of the references I suggested in my original review and it still seems relevant and worth citing.
- Just like SUMS is valuable to search for promiscuous or generally activated variants, it should also be useful for finding more selective variants that convert only certain substrates from mixtures even after long incubations (similar to the case of perfectly enantioselective enzymes that will act on one isomer and not another, even once the preferred isomer is consumed).

Supporting Information

Lines 566-567: I think this is somewhat misleading and would be clearer if "where a product increases in abundance" is changed to "where the relative abundance of a product increases".

Line 574: Again, "the amount of product formed" should probably be "the relative amount of product formed".

Line 576: Here it is correctly described as apparent "increases in relative activity". By keeping this clearly stated, the reader will more easily follow what is said in Lines 577-579.

Line 612: Too late in the document to define the SUMS acronym.

Reviewer #4 (Remarks to the Author):

I am satisfied with the revised manuscript. As was clear from my first review of this manuscript, I was initially sceptical. However, over the past several weeks I have found the concept of this paper frequently coming up while thinking about diverse aspects of high-throughput screening. While seemingly simple at first, I believe that this concept has the potential to significantly increase the amount of information extracted from variant libraries and I still think that perhaps it would be more frequently employed if the approach were more clearly formulated. I think this manuscript makes a good case for the SUMS approach and I hope to see much more of this in the future. I still have a few minor comments that the authors might consider before publication if the manuscript is accepted.

Line 72: "with a peptide mixture that simulating" should be "with a peptide mixture that simulates"

This typo has been corrected.

Lines 180-181: "smaller W349S mutation" sounds strange

The word "smaller" has been removed to simplify the sentence.

Lines 220-221: Be more explicit about how incubation time affects product ratios, perhaps referring to Figure S42.

We have removed this sentence, as the content is somewhat redundant and our effort to provide more detail has instead generated distraction from the main point. All relevant information is still available in the supplemental.

Lines 252-253: The authors could emphasise that this makes screening random mutagenesis libraries rather unrewarding and therefore decreases motivation to do so. If SUMS dramatically increases the chances of finding useful information (e.g., identifying residues that are important to substrate specificity even if overall activity is decreased) then screening random libraries would become much more interesting, even in cases where high-throughput methods are not available.

We thank Reviewer #4 for this suggestion, and we have added further emphasis of the potential for SUMS to identify interesting variants from globally random libraries in the Discussion section of our manuscript.

Line 370: This is a good place to remind the reader that it is usually critical that reactions do not go to completion.

We agree with Reviewer #4 and added a brief clarification to this effect.

Lines 372-374: The authors could emphasise how much more work it would require to get the same information without substrate mixtures (i.e., k_{cat}/K_m would need to be estimated for the individual substrates, merely running the reactions separately would not give the same information).

We have edited this section to provide the recommended emphasis, which has led to smoother and more informative prose.

Lines 398-400: I think an even bigger problem with high-throughput screening is that there is significant inter-well variation in protein expression level, even for the same variant. This means that even if absolute product concentration is accurately quantified, no information on changes in specific activity are obtained. This could be addressed by normalisation strategies using fusion tags for quantification (split-GFP, split-luciferase, S-tag, GST-tag, etc.) but throughput is decreased considerably due to the complexity and time required for these procedures. Worse, some proteins do not tolerate tags at either end, meaning that normalisation in crude lysates is essentially impossible. Overall, this makes high-throughput screening nonquantitative, which in turn severely limits its utility. This problem is making it very difficult to get high-quality data for machine learning projects, to mention just one example. The ability to get useful information on specificity without the need for normalisation based on expression levels is therefore one of the most exciting benefits of the SUMS approach and I believe the authors would make the significance of their work much more obvious by emphasising this aspect.

We are grateful for this insight from Reviewer #4! We describe this exact relationship in the section describing TrpB engineering, but did not repeat it in the discussion. We agree with Rev 4 that further emphasis would be appropriate, and have added a statement reflecting this benefit of SUMS. We have substantiated this topic by including a pair of references to direct readers towards examples of such tagging methods that are used in high-throughput screening.

Line 557: Reference page numbers are wrong: 2193

Line 573: Reference information is wrong: 1999, Pages 1758-1761

Line 575: Page numbers are wrong: 1030

Our references have been edited.

General comments:

- Was Reinen et al., (Anal. Bioanal. Chem., 2016, 408, 1425-1443) left out intentionally? This is one of the references I suggested in my original review and it still seems relevant and worth citing.

While we enjoyed reading this paper, the other multiplexing papers we have cited fall into two main categories: screening on mixtures when the application of an enzyme is in an environment where mixtures are present (i.e., in vivo) OR screening on mixtures when the application is to use enzymes for organic synthesis. Reinen et al. does not fall into either category. Their focus is on characterizing products of drug metabolism, not on substrate specificity. We believe our revised introduction provides a strong overview of relevant examples of substrate multiplexed screening and the addition of this peripherally related topic would prove a distraction. This citation could have a place in a more expansive review of substrate multiplexed screening methods for enzyme characterization.

- Just like SUMS is valuable to search for promiscuous or generally activated variants, it should also be useful for finding more selective variants that convert only certain substrates from mixtures even after long incubations (similar to the case of perfectly enantioselective enzymes that will act on one isomer and not another, even once the preferred isomer is consumed).

We appreciate this comment. While our focus has been on engineering promiscuous biocatalysts, we have added a statement to our Discussion to highlight this alternative application of SUMS.

Supporting Information

Lines 566-567: I think this is somewhat misleading and would be clearer if "where a product increases in abundance" is changed to "where the relative abundance of a product increases".

Line 574: Again, "the amount of product formed" should probably be "the relative amount of product formed".

Line 576: Here it is correctly described as apparent "increases in relative activity". By keeping this clearly stated, the reader will more easily follow what is said in Lines 577-579.

Line 612: Too late in the document to define the SUMS acronym.

We have followed all of the above suggestions.

REVIEWERS' COMMENTS

Reviewer #4 (Remarks to the Author):

Final comment: Page 10, line 225-226: "While the small L355A mutation" sounds strange and this could be edited during final proofing if the paper is accepted.

Response to Reviewer comment

Page 10, line 225-226: "While the small L355A mutation" sounds strange and this could be edited during final proofing if the paper is accepted.

This sentence has been changed to "One such mutation, L355A, improved activity on 4-Br-Trp, but the L355M mutation..."